# Transmissible Cancer Evolution: The Under-Estimated Role of Environmental Factors in the “Perfect Storm” Theory

**DOI:** 10.3390/pathogens11020241

**Published:** 2022-02-12

**Authors:** Sophie Tissot, Anne-Lise Gérard, Justine Boutry, Antoine M. Dujon, Tracey Russel, Hannah Siddle, Aurélie Tasiemski, Jordan Meliani, Rodrigo Hamede, Benjamin Roche, Beata Ujvari, Frédéric Thomas

**Affiliations:** 1CREEC/MIVEGEC, Université de Montpellier, CNRS, IRD, 34394 Montpellier, France; anne-lise.gerard@evobio.eu (A.-L.G.); justine.boutry@ird.fr (J.B.); jordan.meliani@ird.fr (J.M.); benjamin.roche@ird.fr (B.R.); frederic.thomas2@ird.fr (F.T.); 2Centre for Integrative Ecology, School of Life and Environmental Sciences, Deakin University, Waurn Ponds, VIC 32020, Australia; a.dujon@deakin.edu.au (A.M.D.); beata.ujvari@deakin.edu.au (B.U.); 3School of Life and Environmental Sciences, University of Sydney, Sydney, NSW 2006, Australia; tracey.russell@sydney.edu.au; 4School of Biological Sciences, University of Southampton, Southampton SO17 1BJ, UK; H.V.Siddle@soton.ac.uk; 5Institute for Life Sciences, University of Southampton, Southampton SO17 1BJ, UK; 6Université de Lille, CNRS, Inserm, CHU Lille, Institut Pasteur de Lille, U1019-UMR9017-CIIL-Centre d’Infection et d’Immunité de Lille, 59000 Lille, France; aurelie.tasiemski@univ-lille.fr; 7School of Natural Sciences, University of Tasmania, Hobart, TAS 7001, Australia; rodrigo.hamedeross@utas.edu.au; 8Departamento de Etología, Fauna Silvestre y Animales de Laboratorio, Facultad de Medicina Veterinariay Zootecnia, Universidad Nacional Autónoma de México (UNAM), Ciudad de México 01030, Mexico

**Keywords:** transmissible cancers, environmental factors, human activities, transmission, evolution, ecology

## Abstract

Although the true prevalence of transmissible cancers is not known, these atypical malignancies are likely rare in the wild. The reasons behind this rarity are only partially understood, but the “Perfect Storm hypothesis” suggests that transmissible cancers are infrequent because a precise confluence of tumor and host traits is required for their emergence. This explanation is plausible as transmissible cancers, like all emerging pathogens, will need specific biotic and abiotic conditions to be able to not only emerge, but to spread to detectable levels. Because those conditions would be rarely met, transmissible cancers would rarely spread, and thus most of the time disappear, even though they would regularly appear. Thus, further research is needed to identify the most important factors that can facilitate or block the emergence of transmissible cancers and influence their evolution. Such investigations are particularly relevant given that human activities are increasingly encroaching into wild areas, altering ecosystems and their processes, which can influence the conditions needed for the emergence and spread of transmissible cell lines.

The majority of cancers are not transmissible, with several notable exceptions [1]. For example, the transmission of leukemia, melanoma, lymphoma, and carcinoma have been documented in 0.1% of pregnancies in humans, in both mother-to-fetus and fetus-to-fetus transmissions [2,3,4]. A case of cancer transmission from malignant tapeworm cells (*Hymenolepis nana*) to a HIV patient was also recorded in 2015 [5]. In the 1960s, a spontaneous reticulum-cell sarcoma arose in a laboratory colony of Syrian hamsters (*Mesocricetus auratus*) from the National Institutes of Health [6]. Surgical orthotopic implantation of human malignancies into immunodeficient (NOD/SCID) mice is performed routinely in oncological research [7,8,9].

In the wild, there are at least nine known transmissible cancers: one in dogs (Canine Transmissible Venereal Tumor [10]), two in Tasmanian devils (Devil Facial Tumor Disease [11,12]), and six in marine bivalves (Disseminated Neoplasia of Bivalves [13,14]). There is also an atypical case of vertically transmitted tumors in two species of hydra: when tumor-bearing individuals reproduce asexually by budding, tumor cells, and at times tumor-associated bacteria (i.e., in *Hydra oligactis*), are transmitted to the bud which then evolve into tumors in the following weeks [15,16]. Because ecological consequences of transmissible cancers can be dramatic (e.g., [17,18,19]), transmissible cell lines are increasingly considered to be of potential ecological concern. As with other infectious diseases, human activities may disperse these pathogens across the globe, thus reaching pandemic proportions [20,21,22]. However, despite recent advancements in our understanding of the biology of transmissible cancers (see [23] for a recent synthesis), the exact mechanisms of their emergence and evolution remain elusive.

Even if we underestimate the true prevalence of transmissible cancers [24], the consensus currently emerging in the scientific community is that transmissible cancers are equivalent to rare novel parasites. This rarity may appear as a paradox given that transmissible cancers originate from classical cancer cells, and that carcinogenesis occurs in most, if not all, multicellular organisms across the tree of life [25,26]. Because the majority of cancer cells die with their host, transmissible cancers represent a kind of endpoint that is only scarcely crossed in the ongoing process of speciation initiated by tumorigenesis [27,28]. Until now, few studies have attempted to elucidate what keeps cancer cells from evolving to escape from the body and transmit to a new host. According to the “Perfect Storm” theory [1], the emergence of transmissible cancers requires the confluence of multiple tumor and host traits (Figure 1). Firstly, the tumor must shed a substantial number of cells. These cells must remain alive and infective outside of, or on the surface of the original host. When in contact with a new host, they must infect it, evade immune recognition, and be able to proliferate in an appropriate tissue [27]. At the host population level, factors including low genetic diversity [29,30], exposure to environmental stress, and/or parasitic pressures may weaken the host’s immune system and favor the successful establishment and transmission of cancer cells between individuals [31].

Although this confluence of traits is unlikely to frequently occur, the difficulty of acquiring a transmissible capacity remains insufficient to explain the scarcity of transmissible cancers. This is because even events that have a low intrinsic probability of occurrence are likely to yield common phenomena (i.e., because they finally accumulate) when eons of evolution are considered. The rarity of transmissible cancers is likely to be explained by additional environmental factors that prevent the spread and establishment of malignant cell lineages in host populations once the transmissible capacity has been reached.

Predation is likely to be one such effect, which is well-described in the ecology and evolution of parasitic species, but has been overlooked in the case of transmissible cancers [32]. Indeed, a large proportion of parasites never complete their life cycle because they are consumed as prey (either directly as an individual prey item, or indirectly when digested with the body of their host [33]) thus putting those species under selective pressure. The extent to which predation of weak individuals prevents cancers evolving to the transmissible stage is unknown, but this hypothesis is in accordance with several observations. Indeed, it specifically predicts that transmissible cancers should evolve in species that do not have predators (such as Tasmanian devils, see for instance [32]), or those whose health does not influence their predation rate (e.g., cancerous bivalves, when buried in mud or among a bed with healthy individuals, are unlikely to be targeted as preferential prey by birds), or in prey species that are kept in predator-free environments (such as the previously mentioned laboratory hamsters). Hydras provide another relevant example—while tumorous individuals are apparently never observed in the field, certain polyps were observed developing tumors after a few months in a laboratory setting (Boutry and Tissot, unpublished observations). Under experimental conditions, it has been demonstrated that predatory fish strongly preferred tumorous hydras over healthy ones [34]. In this specific case, the higher predation rate seems to result from a higher detection by visual hunting fish, given the altered morphology in tumorous hydras.

Competition (both intra- and inter-specific) and parasitism are likely to be limiting factors in the spread of emerging transmissible cell lines. As soon as hosts harboring transmissible cell lines have a reduced competitive ability, and/or a higher vulnerability to detrimental pathogens, the same consequences as those described for parasitized hosts are expected. In hydras, a tumorous host seems to be more attractive to symbiotic organisms, e.g., ciliates [34], supporting the expectation of a higher vulnerability to pathogens in hosts harboring transmissible cancers. Given that most, if not all, organisms must cope with competitors and parasites in ecosystems, hosts with a depleted body condition, carrying newly emerged transmissible cell lines are likely to die before the transmissible cell lines can spread quickly enough to persist in the population.

Additionally, abiotic environmental variables, such as temperature, PH, oxygen, and CO_2_ levels also deserve consideration. Frequently, parasitized hosts have a modified tolerance to various abiotic environmental variables, and/or extreme values [35,36], compared to healthy ones. The extent to which this is also the case with tumor-bearing individuals has been insufficiently studied to our knowledge but is likely to be relevant given that tumorous hosts share of lot of similarities with parasitized ones [37]. For instance, pollutants could both promote transmissible cancer cells’ emergence (i.e., mutagenic substances), and favor their subsequent spread if they also weaken immunity among host individuals within the population. A similar effect on the spread could be expected if this pollution induces mortality in top predator populations. Conversely, if tumor-bearing individuals are more sensitive to abiotic environmental stressors (i.e., higher mortality rate), the spread of any emerging transmissible cancers cells is likely to be prevented (see also Figure 1).

In conclusion, it seems clear that if the acquisition of a transmissible phenotype by cancer cells is a major limitation for the emergence of transmissible cancers, but environmental variables are also likely to be subsequently a major barrier for the spread of those cells in host populations (Figure 1). From an applied ecological perspective, it also means that we should consider more carefully how anthropic activities are likely to promote the spread of transmissible cancers (see also [20]). When top order predators are removed from ecosystems through human intervention, the survival of weak and sick individuals, that would otherwise be rapidly eliminated, is favored [38]. This enhanced survival of sick individuals increases the time they are able to spread any transmissible entities they carry.

Human activities may also result in an increase in food availability, enhancing the survival of weaker individuals, whether it be by eutrophication of waterways affecting some aquatic species (e.g., [39]), or by the huge quantities of discarded food that we humans trash that becomes accessible to some terrestrial species [40,41]. Another anthropic factor that could play a role in the promotion and emergence of transmissible cancers is the fragmentation of natural habitats which decreases the efficient size of populations and thus their genetic diversity [29,30,42], increasing the probability of transmission between two hosts [43]. Such ecological contexts can, in theory, yield dramatic consequences, even when they are transient because once a threshold has been reached in the population size of the emerging transmissible entities, it can be challenging to remove it even when normal ecological conditions are restored (see [44] for an example of emerging pathogens). Once started, epidemics can be maintained by a vicious circle: as sick individuals die the level of resources available per individual increases allowing the remaining individuals, including infected ones, to extend their survival and the probability of cancer cell transmission [32,45].

We hope that this short perspective will stimulate scientists’ interest in wildlife cancers and pave the way for novel research into the evolutionary enigma of transmissible cancers. We encourage future research to focus on animal species that have, whatever the reasons, a similar ecology or ecological conditions as those of Tasmanian devils. For instance, when anthropogenic changes erase top predators from ecosystems, when increased food resources become available to animals outside of their traditional diets or promote inbreeding, it increases the probabilities that any transmissible cancers can evolve and spread. In a one health perspective, more attention should be given at studying and monitoring such ecological contexts.

## Figures and Tables

**Figure 1 pathogens-11-00241-f001:**
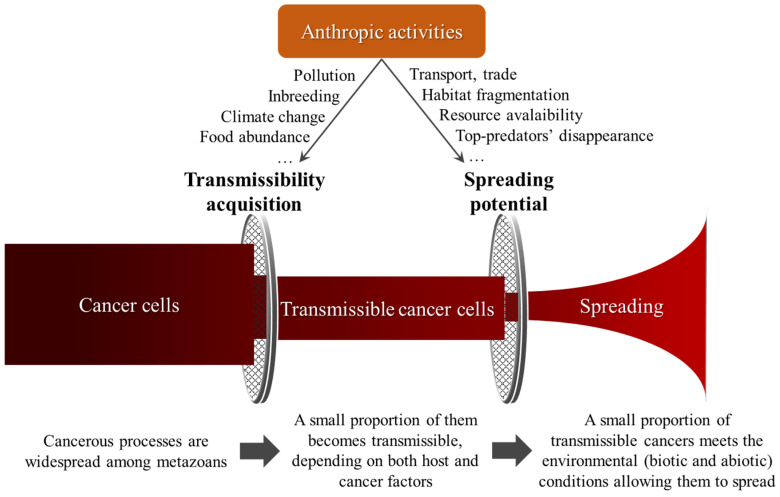
Transmissible cancers depend on a «Perfect Storm» to emerge and to spread. Human activities can influence both components.

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
