# Peer review of "Transmissible Cancer Evolution: The Under-Estimated Role of Environmental Factors in the “Perfect Storm” Theory"

_pathogens, 2022, doi:10.3390/pathogens11020241_

Round 1

Reviewer 1 Report

This article is a perspective on transmissible cancers and why they are not prevalent throughout nature. The perceived rarity of transmissible cancers in nature is peculiar as any rare event is likely to increase in number over large periods of time, and this rarity is instead attributed to both biotic and abiotic factors. The manuscript goes on to provide a brief overview as to these factors that could go some way to explaining the “perfect storm” theory. Overall, the paper is well written and provides a short insight intended to stimulate conversation on transmissible cancer biology.

Legend for Figure one is not spaced out correctly.

The sentence on lines 49-52 “Because ecological consequences of transmissible cancers can be dramatic (e.g. > 85% population decline in 20 years in Tasmanian devils [17] [18]; epizootic outbreaks in marine mollusk populations [19]), transmissible cell lines are increasingly considered to be of potential ecological concern” is clunky and needs to be slightly reworded.

Having read the “Tumors (re)shape biotic interactions within ecosystems: Experimental evidence from the freshwater cnidarian Hydra” paper, it appears that tumor-bearing hydra phenotypically present as being bigger in size and with more tentacles, and they are preferentially favoured by predator fish as such. Do you think the same holds true for other tumor-bearing species with a less obvious tumor phenotype to be preferentially favoured by a predator? Or is it more so to do with the declining health of a tumor-bearing individual that it becomes preferentially targeted?

In the event of an epidemic arising in a species (line 142), maybe make brief mention of the ethical considerations that would need to be taken into account by both the scientific and general community.

Reviewer 2 Report

In the perspective tilted ‘Transmissible Cancer Evolution: the Under-Estimated Role of 2 Environmental Factors in the “Perfect Storm” Theory’ submitted by Tissot et.al, have discussed the prevalence of transmissible cancers that are uncommon in nature. Authors have explained the perfect storm theory hypothesis that suggests that the transmissible cancers are not frequent because a precise convergence of tumor and host traits is needed for them to emerge. Authors have also mentioned that increased human activities may lead to increase the resources such as food availability which in turn enhance the survival of weak/sick individuals and this provides a chance to them to spread any transmissible entities they carry.

Overall, the manuscript is interesting and well written and shall encourage new research in the area.

Reviewer 3 Report

This perpective written by Sophie Tissot et al. highlight the potential roles for environmental factors that should be implemented to the “perfect storm theory” to study the emergence and spread of transmissible cancer lines and consider them for ecological studies of the wildlife.  

This perspective is well written and pleasant to read.

In order to improve this manuscript, I just formulate two suggestions :

  • In the paragraph form line 101 to 105, I think the current available data don’t really allow to draw conclusions about the prevalence of transmissible cancers in top predators. In the case of CTVT for example, that has emerge thousands years ago,  it remains difficult to estimate the predatory pressure the founder hosts were enduring or whether they were top predators them themselves at the emergence time of the CTVT. Additionally, to some extent, Tasmanian devils could be considered as apex predators themselves in their own ecological niche. So I suggest the authors to be more precise and specific in this paragraph.
  • The paragraph about the potential roles of abiotic factors, from line 116 to 121, is of importance as multiple abiotic factors could be involved at two different levels in the emergence of transmissible cancers. Numerous abiotic factors (often pollutants) are known to promote cancerous process, and as the authors mentioned, they also could impact the transmissibility of some cancers. So I suggest to the authors to give more details about the potential roles of abiotic factors in this perfect storm theory.
